# Indirect regulation of HMGB1 release by gasdermin D

Allen Volchuk[1], Anna Ye[1], Leon Chi[1,2], Benjamin E. Steinberg ⓘ [2,3,4,5,6✉] & Neil M. Goldenberg ⓘ [1,2,4,5,6✉]

The protein high-mobility group box 1 (HMGB1) is released into the extracellular space in response to many inflammatory stimuli, where it is a potent signaling molecule. Although research has focused on downstream HMGB1 signaling, the means by which HMGB1 exits the cell is controversial. Here we demonstrate that HMGB1 is not released from bone marrow-derived macrophages (BMDM) after lipopolysaccharide (LPS) treatment. We also explore whether HMGB1 is released via the pore-forming protein gasdermin D after inflammasome activation, as is the case for IL-1β. HMGB1 is only released under conditions that cause cell lysis (pyroptosis). When pyroptosis is prevented, HMGB1 is not released, despite inflammasome activation and IL-1β secretion. During endotoxemia, gasdermin D knockout mice secrete HMGB1 normally, yet secretion of IL-1β is completely blocked. Together, these data demonstrate that in vitro HMGB1 release after inflammasome activation occurs after cellular rupture, which is probably inflammasome-independent in vivo.

[1] Program in Cell Biology, Hospital for Sick Children, 686 Bay St, Toronto, ON M5G 0A4, Canada. [2] Department of Physiology, The University of Toronto, 1 King's College Circle, Toronto, ON M5S 1A8, Canada. [3] Program in Neuroscience and Mental Health, Hospital for Sick Children, 686 Bay St, Toronto, ON M5G 0A4, Canada. [4] Department of Anesthesia and Pain Medicine, Hospital for Sick Children, 555 University Ave, Toronto, ON M5G 1X8, Canada. [5] Department of Anesthesiology, The University of Toronto, 123 Edward St, Toronto, ON M5G 1E2, Canada. [6]These authors jointly supervised this work: Benjamin E. Steinberg, Neil M. Goldenberg. ✉email: benjamin.steinberg@sickkids.ca; neil.goldenberg@sickkids.ca

H igh-mobility group box 1 (HMGB1) is a ubiquitously expressed protein that can be released from cells during infection or sterile inflammation[1]. HMGB1 can exist in several spatial pools, with the bulk of HMGB1 in a resting cell residing in the nucleus. HMGB1 weakly binds to chromosomal DNA and can influence DNA transcription through the regulation of chromatin structure[2]. Indeed, loss of HMGB1 from macrophages results in nuclear reprogramming toward a more activated state[3]. Multiple stimuli can lead to HMGB1 translocation to a cytosolic pool[1]. This is thought to result from acetylation of two nuclear localization sequences on HMGB1[4]. In addition, phosphorylation of cytosolic HMGB1 can inhibit its import to the nucleus, shifting its steady state toward the cytosol[5]. Bacterial products, such as lipopolysaccharide (LPS), and tissue injury can stimulate the release of HMGB1 from both immune and parenchymal cells[1,6]. Once in the extracellular space, HMGB1 is a potent signaling molecule. The binding of HMGB1 to its cognate receptors—including the receptor for advanced end-glycation products, toll-like receptor 4, and others—stimulates cytokine secretion from macrophages, and can also promote proliferation, migration, and other phenotypic changes in somatic cells[7–9]. Its pleiotropic effects make extracellular HMGB1 the focus of translational research in a wide variety of fields ranging from sepsis, acute respiratory distress syndrome, inflammatory arthritis, and pulmonary vascular diseases[1,6,10].

While the importance of extracellular HMGB1 is well documented, precisely how HMGB1 is released from cells remains poorly understood. Mechanistic knowledge of the secretory route taken by HMGB1 would open the door for therapeutic manipulation of HMGB1 release, with clear implications for inflammatory and infectious diseases.

Any form of lytic cell death results in passive HMGB1 release concurrent with all cellular contents[1,11]. On the other hand, active, programmed secretion pathways for HMGB1 release have long been proposed but remain less understood. Two models in particular are most frequently cited. In one, stimulation of macrophages with LPS results in bulk translocation of HMGB1 from the nucleus to the cytosol, with associated secretion of HMGB1 into the extracellular space[4,12]. While the movement of HMGB1 from nucleus to cytosol is thought to arise from acetylation of nuclear localization sequences on HMGB1[4], the subsequent secretion of HMGB1 is not explained. Such a mechanism would need to account for the secretion and/or packaging of a soluble protein lacking any relevant signal sequence. To that end, the second frequently cited route for regulated HMGB1 secretion involves the packaging of HMGB1 into intracellular vesicles, perhaps of lysosomal or autophagosomal origin, and subsequent release as these vesicles through fusion with the plasma membrane. This model also has unexplained steps, including the signal governing HMGB1 packaging and how HMGB1 would resist proteolysis once inside a lysosome. HMGB1 release from cells has been shown to occur as a result of the programmed cell death pathways ferroptosis[13] and necroptosis[14]. Of note, HMGB1 does not appear to be released during apoptosis, even in the presence of strong detergents[15].

The relatively recent discovery of the inflammasome has provided yet another potential route for HMGB1 secretion[16,17]. These multi-protein complexes allow for the release of the soluble cytokines interleukin (IL)-1α, IL-1β, and IL-18 into the extracellular space. The best described inflammasome is the NLRP3 complex, which can be activated by bacterial products, ATP, crystals, or other endogenous indicators of cellular damage[17]. Its activation ultimately leads to cleavage of the pro-form of the above cytokines along with another protein substrate called gasdermin D (GSDMD) through an activated caspase-1[18–21]. The N-termini of cleaved GSDMD oligomerize within the plasma membrane to form large pores through which the mature cytokines are secreted[22]. Under some conditions, this is followed by cell rupture, termed pyroptosis, although it is now known that inflammasome activation need not result in cell death[18,23,24]. The existence of this large pore has caused others to speculate that HMGB1 may be secreted through GSDMD, although this has also not been definitely proven[25].

Herein, we analyze the route by which HMGB1 is released by macrophages, assessing each of the hypothetical models detailed above. We show that HMGB1 is not released following LPS stimulation alone nor packaged into vesicles. While inflammasome activation does indeed lead to HMGB1 release in vitro in a GSDMD-dependent fashion, HMGB1 does not appear to exit the cells through the pore. Instead, HMGB1 is released to the medium following pyroptosis-induced cellular lysis. Finally, in a mouse model of endotoxemia, we show that loss of the GSDMD pore prevents IL-1β secretion into the blood while having no effect on HMGB1 release, demonstrating that GSDMD-mediated secretion is not a significant source of plasma HMGB1 during systemic endotoxemia. Together our data argue against a vesicle- or directly GSDMD-mediated release of HMGB1.

## Results

**LPS does not stimulate HMGB1 release from macrophages.** The literature contains conflicting reports regarding the secretion of HMGB1 in response to LPS alone, with some papers demonstrating this effect[4], and others that do not[26]. To evaluate whether LPS alone leads to HMGB1 secretion, we first assessed HMGB1 secretion in bone marrow-derived macrophages (BMDM) by ELISA. LPS-stimulated BMDM did not release HMGB1 into the extracellular medium (Fig. 1a). In addition, LPS stimulation did not change HMGB1 expression levels (Fig. 1b). Moreover, immunostaining for HMGB1 in LPS-stimulated cells did not reveal any translocation from the nucleus to an intracellular vesicular compartment (Fig. 1c, d), such as the endolysosome compartment. Specifically, HMGB1 did not colocalize with either Rab7 or LAMP1, respective markers of late endosomes and lysosomes (Fig. 1c, d). While some reports have documented the bulk translocation of HMGB1 from nucleus to cytosol following LPS stimulation, we do not observe this phenomenon. The lack of observable HMGB1 puncta in LPS-stimulated BMDM similarly argues against any other secretory vesicle-mediated HMGB1 secretion pathway.

**Pyroptosis activators lead to HMGB1 release from macrophages.** Having failed to elicit HMGB1 release using LPS alone, we next sought to test for inflammasome-mediated HMGB1 release by stimulating mouse BMDM with known activators of pyroptosis. In LPS-primed BMDM, nigericin or potassium depletion activated the inflammasome as confirmed by the presence of mature IL-1β in the extracellular culture medium, detected by immunoblotting (Fig. 2a). No IL-1β secretion was observed in untreated cells or in cells treated with LPS alone. HMGB1 was found in the culture medium under the same inflammasome-activating conditions (Fig. 2a). GAPDH, present in the cytosol as a tetramer of 37 kDa subunits[27], was also found in the extracellular medium in all conditions of inflammasome activation (Fig. 2a). Since this protein complex is larger than the GSDMD pore, we hypothesized that this was indicative of cellular lysis during inflammasome activation. To further investigate this finding, we performed a colorimetric assay to test for LDH release from the BMDM under the same set of conditions. Release of LDH, a large cytosolic protein, is a common indicator of cell rupture and cytotoxicity[24,28]. Consistent with prior publications, LPS + nigericin or potassium depletion caused pyroptosis, as

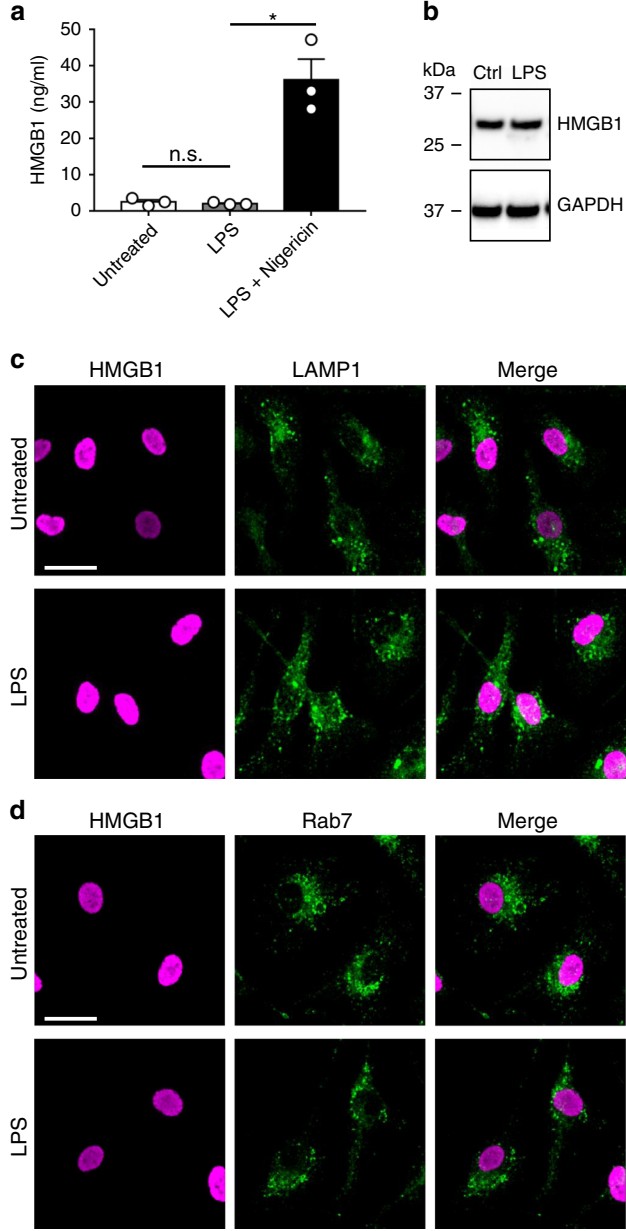

**Fig. 1 LPS treatment alone does not lead to HMGB1 secretion in BMDM. a** BMDM were untreated or primed with LPS for 5 h (0.5 μg/ml) and then treated with or without nigericin (Ng; 20 μM) for the last 30 min. HMGB1 present in the extracellular media (supernatant) was quantified by ELISA. $n = 3$ (mean ± SEM). LPS treatment alone did not result in secretion of HMGB1 into the extracellular media. *$p = 0.0009$, n.s. not significant as determined by ANOVA with Tukey's multiple comparison correction, two-sided. **b** BMDM were untreated, or treated with LPS for 5 h (0.5 μg ml⁻¹). Cells were lysed and immunoblotted for HMGB1 and GAPDH. Note that HMGB1 expression does not change in response to LPS treatment. **c**, **d** BMDM were primed with LPS for 5 h (0.5 μg ml⁻¹). The cells were fixed and immunostained for HMGB1 and LAMP1 or Rab7 and imaged by spinning disk confocal microscopy. Collapsed (Extended focus) images at ×63 magnification, size bar 20 μm. HMGB1 and LAMP1 or Rab7 do not colocalize in unstimulated or LPS-treated cells.

indicated by LDH release (Fig. 2b). In comparison, LPS treatment alone caused minimal release of LDH, comparable to that of control untreated cells (Fig. 2b). Thus, the appearance of LDH in the supernatant mimics the release of GAPDH detected by western blotting.

To confirm the link between inflammasome activation and HMGB1 release, we performed immunofluorescence on BMDM treated with LPS alone or in combination with nigericin. LPS + nigericin stimulated inflammasome activation in a large proportion of BMDM, as shown by the presence of ASC specks (Fig. 2c). In cells with ASC specks, HMGB1 was lost almost entirely from the cell, indicating a strong relationship between inflammasome activation and HMGB1 release. From these initial data, two possibilities exist: HMGB1 is either released by viable BMDM or escapes from BMDM following lytic cell death.

**Pyroptosis-induced HMGB1 release is gasdermin D-dependent.** The data presented above indicate that inflammasome activation results in concomitant release of IL-1β and HMGB1 into the extracellular space, and that cytotoxicity occurs simultaneously with this release. Since other cytosolic proteins, including IL-1β and IL-18, are secreted from cells through GSDMD, we hypothesized that HMGB1 may traverse a similar route. To determine if the GSDMD pore might be required for HMGB1 release in response to LPS + nigericin we first inhibited GSDMD with necrosulfonamide[21]. Extracellular release of both IL-1β and HMGB1 was completely inhibited by necrosulfonamide (Fig. 3a), as was cytotoxicity (LDH release) (Fig. 3b). As a more specific approach to inhibit the GSDMD pore, BMDM were isolated from $Gsdmd^{-/-}$ mice or littermate controls as described in Materials and Methods. The lack of GSDMD protein in these cells was confirmed by western blot (Fig. 3d). Following treatment with LPS + nigericin, $Gsdmd^{-/-}$ BMDM did not undergo pyroptosis, as indicated by the lack of LDH release as compared to littermate control BMDM (Fig. 3c). As expected, LPS + nigericin treatment of wild-type cells stimulated the release of HMGB1 and processed IL-1β into the extracellular medium (Fig. 3e). However, in the absence of GSDMD, none of LDH, IL-1β, nor HMGB1 are released from the BMDM following inflammasome activation, despite the fact that the inflammasome was activated even in $Gsdmd^{-/-}$ BMDM (ASC specks in Fig. 3f, caspase 1 cleavage Supplementary Fig. S1). Therefore, as with IL-1β, inflammasome-mediated release of HMGB1 is GSDMD dependent.

**HMGB1 is not secreted through the gasdermin D pore.** Two scenarios still remain consistent with these data. In one, HMGB1 directly traverses the GSDMD pore in a manner similar to IL-1β. In a second model, GSDMD inserts in the membrane, and under in vitro conditions, water then enters the cell, resulting in cell lysis. HMGB1 then exits the cell in a nonspecific fashion, thereby relying on GSDMD as a stimulus for cell rupture, not as a direct conduit for HMGB1. This distinction is critical since it has recently been demonstrated that inflammasome activation need not result in pyroptosis[18,24,29]. In an effort to separate inflammasome activation from pyroptosis and cellular rupture, we repeated several experiments in the presence of glycine. While its mechanism of action is poorly characterized, glycine is known to prevent pyroptotic cell lysis, while leaving the upstream steps of inflammasome activation and insertion of GSDMD into the plasma membrane unperturbed[30]. In line with this prior observation, the addition of glycine effectively blocks LDH release in response to LPS + nigericin or potassium depletion (Fig. 4a) without affecting the secretion of IL-1β (Fig. 4b, c). However, glycine inhibits the release of HMGB1 from BMDM (Fig. 4b, d), consistent with the notion that that HMGB1 release in response to inflammasome activation occurs only in the presence of lytic cell death.

To further investigate this finding, we employed two other reagents known to activate inflammasomes while resulting in minimal cell death. One reagent tested was 1-palmitoyl-2-

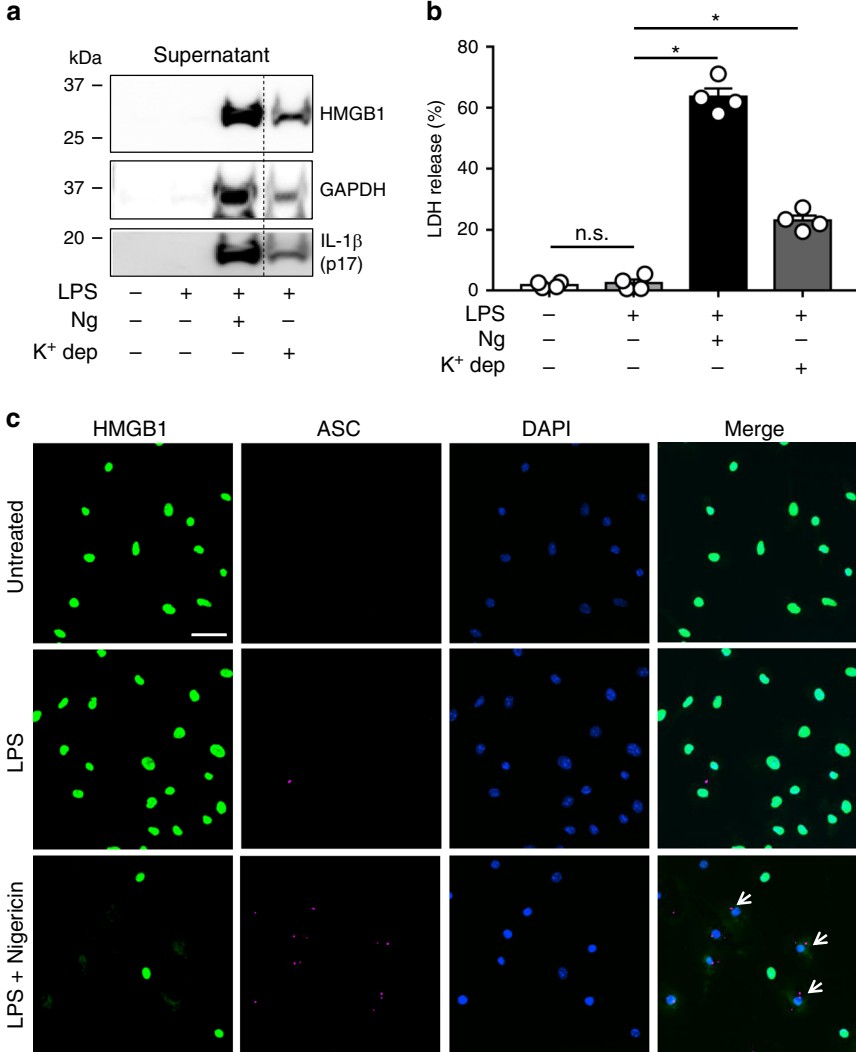

**Fig. 2 Pyroptosis activators lead to HMGB1 release from macrophages.** BMDM were primed with LPS (0.5 µg ml⁻¹ for 4.5 h) and then treated with nigericin (Ng; 20 µM for 30 min) or potassium depletion (K⁺ dep; 2 h) as described in "Methods". Untreated controls were not primed with LPS. **a** Immunoblot analysis of released HMGB1, GAPDH, and cleaved IL-1β (p17) in the cell culture media (supernatant). Dashed line indicates removed lanes. Both LPS + Ng and LPS + K⁺ dep result in the release of HMGB1, GAPDH, and processed IL-1β into the supernatant. Data depict $n = 3$ independent experiments. **b** LDH present in the extracellular media (supernatant) was quantified as a measure of pyroptosis-induced cytotoxicity. Data are expressed as supernatant LDH as a % of total LDH from lysates and supernatants, from $n = 4$ independent experiments. **c** Immunofluorescence analysis of HMGB1 (green) and ASC (red) in BMDM treated as indicated. ASC oligomerized into inflammasome specks (white arrows) in LPS + Ng treated cells. DAPI was used to identify cell nuclei; size bar 20 µm. Data with error bars are represented as mean ± SEM. Each panel is a representative experiment of at least three independent experiments. $*p < 0.0001$ and n.s. not significant as determined by ANOVA with Tukey's multiple comparison correction, two-sided.

glutaryl-*sn*-glycero-3-phosphocholine (PGPC), which is a component of oxidized phospholipids[31]. These lipids, released by dead cells, are known to induce inflammasome activation and IL-1β release in the absence of cell death[18]. The other reagent which induces IL-1β release from living BMDM is a mutant strain of S. aureus lacking O-acetyltransferase A (ΔOatA)[32]. We first confirmed that both reagents cause minimal cell lysis and LDH release in BMDM pre-treated with LPS (Fig. 4e). Consistent with our other findings, both PGPC and ΔOatA stimulated IL-1β processing and release in LPS-treated BMDM (Fig. 4f). As seen when cell lysis was prevented with glycine, HMGB1 was not significantly released in response to either of these reagents as measured by ELISA (Fig. 4g) or HMGB1 localization, which remained nuclear (Fig. 4h, i). Taken together, our data show that inflammasome activation is not sufficient for HMGB1 release, and that unlike prior descriptions[26,33], HMGB1 release in this context of inflammasome activation requires lytic cell death or membrane rupture.

**HMGB1 is released through other large membrane pores**. Our data suggest that HMGB1, under conditions of inflammasome activation, is not released through GSDMD pores, but is released passively during cell lysis. To further investigate these findings, we assessed the effect of another membrane pore on HMGB1 release from BMDM. *S. pneumoniae* produces a pore-forming toxin called pneumolysin[34]. Pneumolysin inserts in the plasma membrane of host cells, and forms a large pore, with an internal diameter of ~30 nm[35], which is 2–3 times larger than the size of the GSDMD pore. Upon treatment of BMDM with pneumolysin, even in the absence of LPS priming, there is significant release of both HMGB1 and LDH into the culture medium (Fig. 5a, b). In contrast, pneumolysin has no effect on IL-1β, release, which

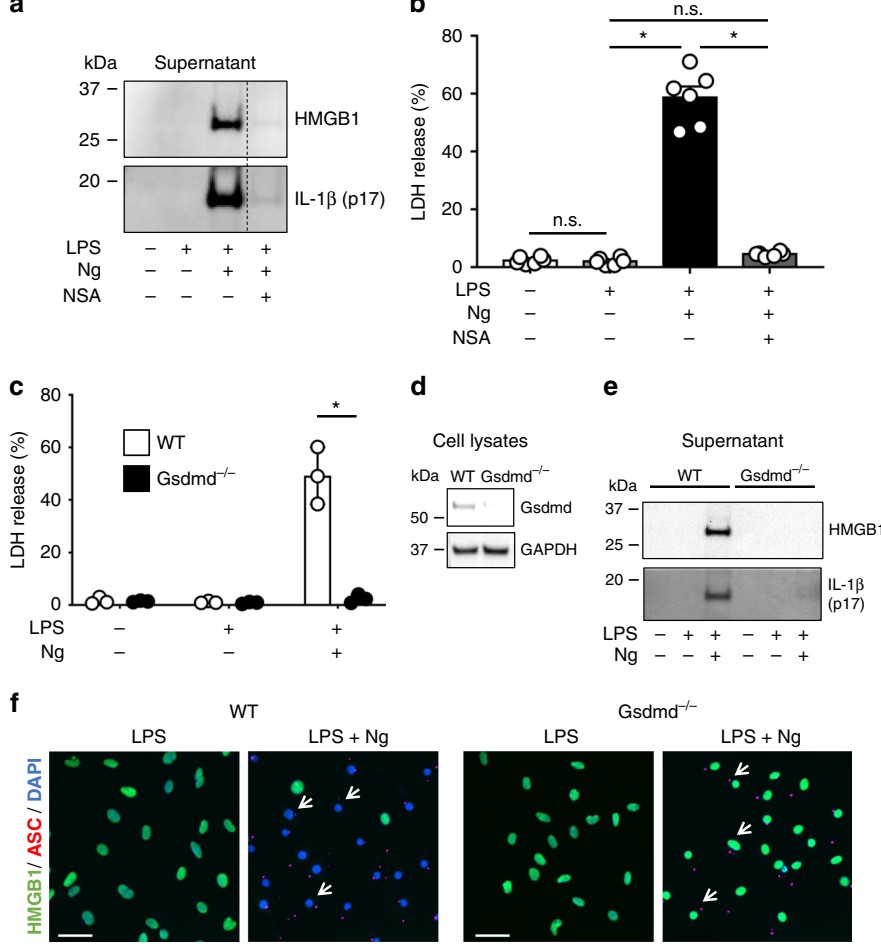

**Fig. 3 Pyroptosis-induced HMGB1 release is gasdermin D-dependent.** BMDM prepared from wild-type mice were primed with LPS (0.5 μg ml$^{-1}$) for 4.5 h, then treated with or without nigericin (Ng; 20 μM) for 30 min. Necrosulfonamide (NSA) was added 3 h after LPS addition as indicated. Immunoblot analysis was used to detect released HMGB1 and cleaved IL-1β (p17) into the cell culture supernatant (**a**). LDH present in the extracellular media (supernatant) was quantified as a measure of pyroptosis-induced cytotoxicity. Data shown from $n = 6$ independent experiments (**b**). NSA inhibits HMGB1 and IL-1β release, as well as cytotoxicity. BMDM prepared from wild-type mice or gasdermin D knockout (*Gsdmd*$^{-/-}$) BMDM were primed with LPS for 4.5 h, then treated with nigericin (Ng; 20 μM) for 30 min. NSA was added 3 h after LPS addition as indicated. LDH present in the extracellular media was quantified as a measure of pyroptosis-induced cytotoxicity (**c**). Data in **c** indicate $n = 3$ independent experiments. Immunoblot analysis of released HMGB1 and cleaved IL-1β (p17) into the cell culture supernatant, showing the loss of HMGB1 or IL-1β release in response to LPS + Ng in *Gsdmd*$^{-/-}$ BMDM (**e**). BMDM lysates from WT and *Gsdmd*$^{-/-}$ animals were blotted for gasdermin D with GAPDH as a loading control (**d**). Immunofluorescence of WT and *Gsdmd*$^{-/-}$ (**f**). BMDM primed with LPS and treated with or without nigericin (Ng). Green and red channels correspond to HMGB1 and red ASC oligomerized into inflammasome specks (white arrows), respectively. DAPI was used to identify cell nuclei. Note that ASC positive *Gsdmd*$^{-/-}$ cells retain nuclear HMGB1. Data with error bars are represented as mean ± SEM. Each panel is a representative experiment of at least three independent experiments. *$p < 0.0001$, n.s. not significant as determined by ANOVA with Tukey's multiple comparison correction, two-sided.

requires LPS priming and a pyroptosis activator (e.g., nigericin or potassium depletion). As expected, pneumolysin alone causes a large reduction in nuclear HMGB1 levels (Fig. 5c).

These data, as well as our data shown above regarding HMGB1 release in response to LPS + nigericin, are surprising given the overwhelming localization of HMGB1 to the nucleus in resting cells. To our knowledge, neither nigericin nor pneumolysin directly affect the nuclear envelope, and therefore should not increase nuclear permeability for HMGB1. Together, these data demonstrate that a large plasma membrane pore is sufficient for the release of HMGB1 from cells. Additionally, these data are consistent with a model whereby HMGB1 is in rapid dynamic equilibrium between the nucleus and cytosol, with the vast majority of HMGB1 inside the nucleus at steady state in intact cells.

**LPS increases serum HMGB1 independent of gasdermin D.** The release of HMGB1 into the blood during endotoxemia has been well-documented[9,10,36]. Endotoxemia has similarly been shown to stimulate inflammasome activation[26] and, GSDMD deletion reduces mortality in this model[20,21]. Given that (a) endotoxemia stimulates inflammasomes, (b) GSDMD blockade is protective, and (c) HMGB1 is released during endotoxemia, several groups have drawn the conclusion that HMGB1, therefore, is released in vivo by an inflammasome- and GSDMD-dependent mechanism[25]. However, the precise relationship of these states has not been thoroughly probed, and an alternative hypothesis is conceivable, whereby HMGB1 release and inflammasome activation are, in fact, parallel processes. Our in vitro data demonstrate that inflammasome-mediated HMGB1 release requires cell lysis, and there is a growing body of evidence

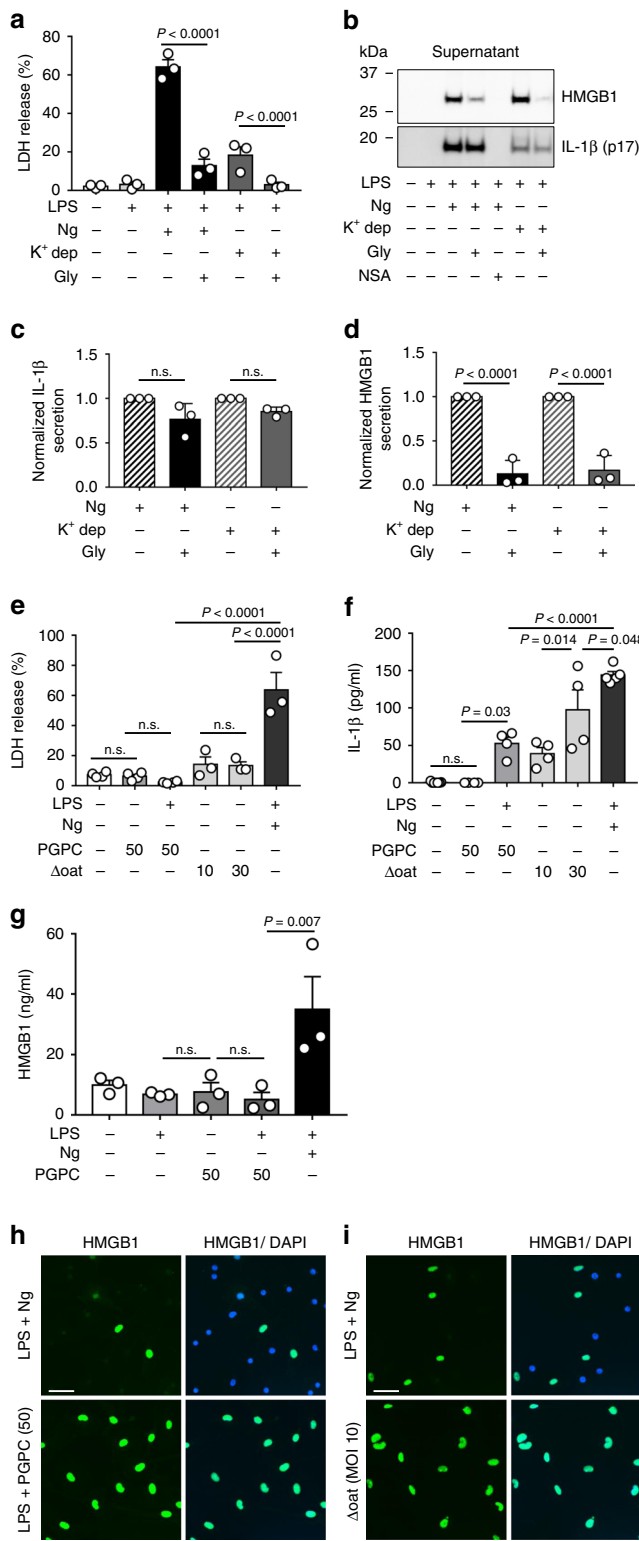

**Fig. 4 HMGB1 is not secreted through the gasdermin D pore.** BMDM were primed with LPS ($0.5\ \mu g\ ml^{-1}$; 4.5 h) and then treated with nigericin (Ng; 20 μM for 30 min) or potassium depletion (K⁺ dep; 2 h) in the absence or presence of 5 mM glycine as described in "Methods". Untreated controls were not primed with LPS. **a** LDH present in the extracellular media (supernatant) was quantified as a measure of pyroptosis-induced cytotoxicity. Glycine effectively prevents cytotoxicity induced by LPS + Ng or K⁺ dep. Data shown from $n = 3$ independent experiments. **b** Immunoblot analysis of released HMGB1 and cleaved IL-1β (p17) into the cell culture media (supernatant) under the conditions tested in **a**. Glycine decreases HMGB1 release without significantly affecting IL-1β secretion. Quantitation of Western blots in **b**. Data in **c** and **d** shown from $n = 3$ independent experiments. BMDM were treated with or without LPS, then treated with PGPC ($50\ \mu g\ ml^{-1}$) for 4 h, or were infected with Δoat S. aureus for 18 h at the MOI indicated, as described in the Methods. Cells treated with Δoat S. aureus were not primed with LPS. **e** LDH release under the indicated conditions. Data in **e** show $n = 4$ (untreated, PGPC, PGPC + LPS) or $n = 3$ (ΔOAT10, ΔOAT30, LPS + Ng) independent experiments. IL-1β (**f**) and HMGB1 (**g**) present in the extracellular media were quantified by ELISA. Both PGPC and Δoat S. aureus stimulate IL-1β release, but not HMGB1 secretion. Data shown in **f** from untreated ($n = 7$), PGPC, LPS + PGPC, ΔOAT10, ΔOAT30 (all $n = 4$), LPS + Ng ($n = 5$) independent experiments. Data shown in **g** from $n = 3$ independent experiments. **h, i** Immunofluorescence of HMGB1 and DAPI, indicating nuclear retention of HMGB1 under the above conditions, and loss of HMGB1 following LPS + Ng. Data with error bars are represented as mean ± SEM. Each panel is a representative experiment of at least three independent experiments. Adjusted p values, provided in the panels, and n.s. not significant as determined by ANOVA with Tukey's multiple comparison correction, two-sided.

hepatocyte source[37] and remain elevated along with other pro-inflammatory cytokines including IL-1β within hours of LPS administration[36]. Mice were injected with 20 mg kg⁻¹ LPS or vehicle, and were sacrificed 6 h later. Blood was collected by cardiac puncture. Splenic whole cell lysates (Fig. 6a, b) demonstrate LPS-dependent upregulation of NLRP3 and proteolytic cleavage of caspase-1, indicative of inflammasome activation. As has been shown previously, wild-type animals secreted both HMGB1 and IL-1β into the blood following LPS treatment (Fig. 6c, d). As expected, $Gsdmd^{-/-}$ animals did not secrete IL-1β in response to LPS, demonstrating the requirement for GSDMD in IL-1β secretion. In contrast, HMGB1 release into the blood in response to LPS was not affected by GSDMD loss. Both wild-type and $Gsdmd^{-/-}$ animals secreted similar amounts of HMGB1, in spite of the fact that $Gsdmd^{-/-}$ mice did not secrete IL-1β, and had similar degrees of inflammasome activation as wild-type mice.

Taken together, these data demonstrate several novel phenomena. First, HMGB1 release in vivo does not require GSDMD. Second, IL-1β and HMGB1 do not exit cells by the same pathway in this model. Third, given that inflammasome activation in vitro leads to IL-1β release in all cases, yet only stimulates HMGB1 release if cells rupture, our in vivo data are consistent with a model in which cell lysis (pyroptosis) is not a significant feature of inflammasome activation in vivo.

## Discussion

Inflammasome activation is a common feature of many infectious and inflammatory states, and has become a focus of intensive research in the fields of innate immunity and cell death[36]. Inflammasomes can propagate inflammatory responses in at least two ways: First, by stimulating the processing and release of cytokines, and second, by causing cell rupture (pyroptosis) and

suggesting that lytic cell death may not be a common endpoint after inflammasome activation in vivo. Indeed, macrophage can activate inflammasomes without cell lysis[18], and cells can repair membrane pores using the ESCRT III system and survive following inflammasome activation[24].

To test the hypothesis that HMGB1 release and pyroptotic cell death are spatially unrelated in endotoxemia, we used an acute intraperitoneal LPS injection model in wild-type and $Gsdmd^{-/-}$ mice. In this model, serum HMGB1 levels increase likely from a

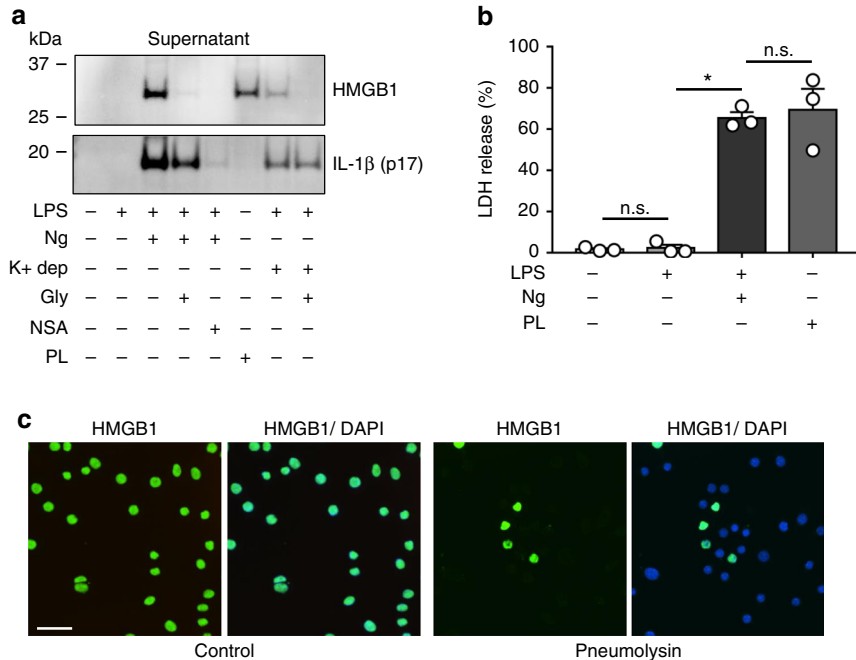

**Fig. 5 HMGB1 is released through sufficiently large membrane pores.** BMDM were primed with LPS (0.5 µg ml$^{-1}$; 4.5 h) and then treated with nigericin (Ng; 20 µM for 30 min), potassium depletion (K$^+$ dep; 2 h) or treated with pneumolysin (PL) alone. Untreated controls and pneumolysin-treated cells were not primed with LPS. Cells were treated with or without 5 mM glycine (Gly), necrosulfonamide (NSA) (5 µM), or PL (0.5 µg ml$^{-1}$). **a** Immunoblot analysis of released HMGB1 and cleaved IL-1β (p17) into the cell culture supernatant. PL causes release of HMGB1, but not IL-1 β processing. **b** LDH present in the extracellular media was quantified as a measure of cytotoxicity, with LPS + Ng and PL causing similar amounts of cell lysis. Data shown from n = 3 independent experiments. **c** Immunofluorescence of control and PL-treated BMDM. HMGB1 (green) and nuclei (DAPI; blue) indicate loss of nuclear HMGB1 in response to PL treatment. Data with error bars are represented as mean ± SEM. Each panel is a representative experiment of at least three independent experiments. *$p < 0.0001$ and n.s. not significant as determined by ANOVA with Tukey's multiple comparison correction, two-sided.

subsequent release of intracellular danger signals[23,38]. The identification of GSDMD solved a long-standing question regarding the release of IL-1β, which is transcribed without a signal sequence, and therefore is not packaged into secretory vesicles[19,22]. The identification of a pore-forming protein, cleaved and activated by the same proteases as those that cleave pro-IL-1β, provided an elegant mechanism for the regulated secretion of a soluble cytosolic cytokine.

HMGB1, an intracellular damage-associated molecular pattern (DAMP), faces a similar topological issue if it is to be actively secreted from immune cells. Namely, it is a soluble protein, largely residing in the nucleus, with no signal sequence or other means of being packaged with normal secretory cargoes. In a highly cited early study, it was proposed that HMGB1 leaves the cell via secretory lysosomes[39]. In our studies, we found no evidence for translocation of HMGB1 into the endolysosomal compartment. Moreover, we did not observe significant redistribution of HMGB1 from the nucleus to the cytosol and/or secretion to the extracellular medium in LPS-stimulated macrophages. This contrasts with other groups that have found HMGB1 in the extracellular medium following LPS treatment alone[33,40,41]. In these studies, wherever cytotoxicity was assessed, the conditions leading to HMGB1 release generated significant cell lysis. Therefore, it is plausible that cell lysis, and not regulated secretion, are the means of HMGB1 release in these previous studies.

The mechanism by which HMGB1 is released into the circulation is important given the known pleiotropic physiologic effects of circulating HMGB1, and its potential as a biomarker and therapeutic target in disease[6]. Because inflammasome activation has been previously shown to induce HMGB1 release, it was tempting to hypothesize that HMGB1, like IL-1β, traverses the GSDMD pore to be secreted[25]. Herein, our studies directly address the role of GSDMD in HMGB1 release. We found that HMGB1 does not appear to move through the GSDMD pore, but GSDMD-dependent, pyroptosis-induced, cell lysis is sufficient for HMGB1 release into the extracellular space. We demonstrate in vitro that inflammasome activation only causes HMGB1 secretion in the context of cell lysis. Prevention of lysis with glycine, or use of non-lytic inflammasome activators, efficiently block the release of HMGB1 into the extracellular space. Furthermore, we show that insertion of very large pores into the membrane (created by pneumolysin) is sufficient for the rapid release of HMGB1, together with cellular rupture. While GSDMD pores are theoretically large enough to allow HMGB1 egress, HMGB1 may exist in a multimeric state and/or bound to other proteins or nucleic acids, largely increasing its effective hydrodynamic radius. Of note, it remains possible that longer durations of LPS exposure, both in vitro and in vivo, may reveal HMGB1 release through the GSDMD pore. This is especially interesting given the potential role for caspase 11-dependent, non-canonical inflammasome-mediated routes of secretion. As an intracellular LPS sensor, this complex may be involved in directing HMGB1 secretion over longer periods of LPS priming, and will be the subject of future investigations[42].

Together, these in vitro data suggest that when the NLRP3 inflammasome is activated in macrophages, HMGB1 release is GSDMD dependent but does not traverse the GSDMD pore. Instead, under these conditions, HMGB1 release requires pyroptotic cell death. Notably, our results do not preclude other active HMGB1 secretion pathways. For example, cellular oxidative stress drives Atg5-dependent autophagy-mediated HMGB1 release[43,44]. This unconventional secretion pathway is notable in that in a reciprocal relationship, cytosolic HMGB1 has been

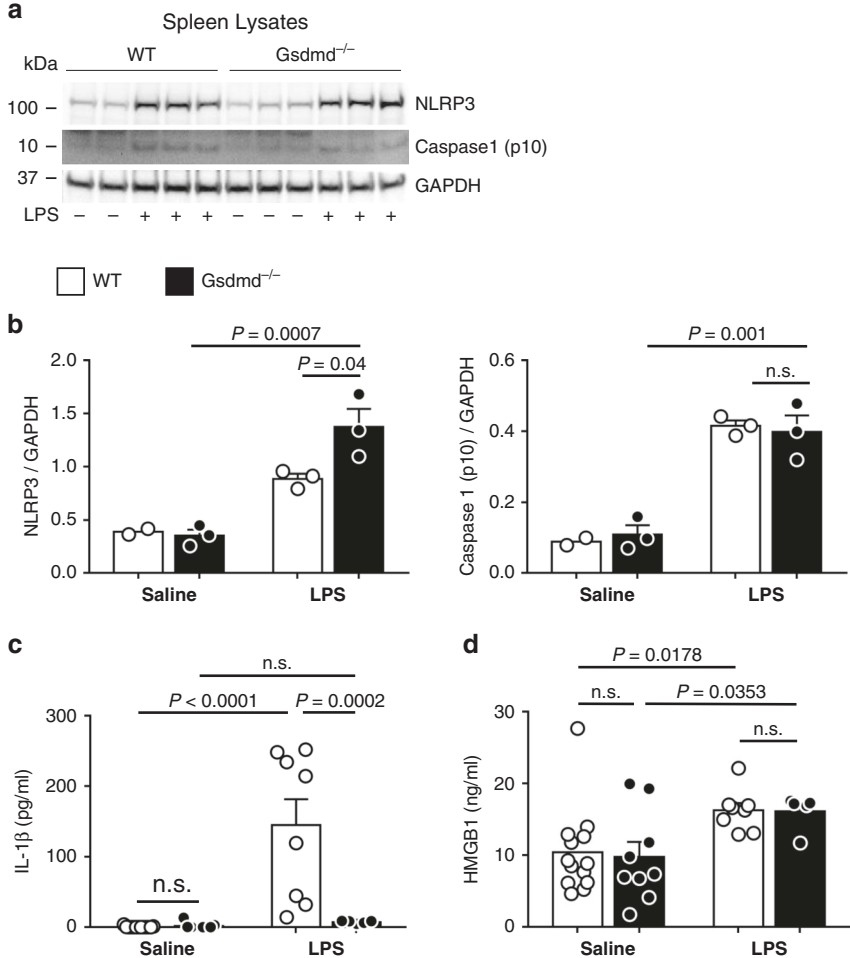

**Fig. 6 Endotoxemia leads to elevated plasma HMGB1 in the absence of gasdermin D.** Wild-type (WT) or gasdermin D knockout ($Gsdmd^{-/-}$) mice were treated with 20 mg kg$^{-1}$ of LPS i.p. or saline. After 6 h, plasma and spleens were collected for analysis. **a** Spleen lysates were immunoblotted for NLRP3 and cleaved caspase-1 (p10) to assess for inflammasome activation, and quantified in **b**. Data indicate WT saline ($n = 2$), WT LPS ($n = 3$), $Gsdmd^{-/-}$ saline ($n = 3$), $Gsdmd^{-/-}$ LPS ($n = 3$) animals per group. NLRP3 upregulation and caspase-1 cleavage were similar in both LPS-treated WT and $Gsdmd^{-/-}$ mice. Plasma from each mouse was collected and analyzed for **c** IL-1β and **d** HMGB1 by ELISA. For **c** and **d**, data indicate WT saline ($n = 13$), WT LPS ($n = 8$), $Gsdmd^{-/-}$ saline ($n = 9$), $Gsdmd^{-/-}$ LPS ($n = 5$) mice per group. Note that $Gsdmd^{-/-}$ mice do not secrete IL-1β in response to LPS, but HMGB1 secretion is unchanged. Data with error bars are represented as mean ± SEM. Adjusted $p$ values, provided in the panels, and n.s. not significant as determined by two-way ANOVA with Tukey's multiple comparison correction, two-sided.

shown to be protective against endotoxemia and bacterial infection via its interaction with the autophagy pathway[3]. By activating autophagy, HMGB1 inhibits inflammasomes, consistent with reports that basal autophagy can suppress IL-1β secretion.

The distinction between direct egress through the GSDMD pores versus cell membrane rupture has important implications to in vivo HMGB1 physiology. Specifically, terminal pyroptotic cell lysis following inflammasome activation remains a topic of debate. Initial studies in pyroptosis, conducted in vitro and often using chemical stimuli such as nigericin in LPS-primed cells, demonstrated a high rate of cell rupture following inflammasome activation[40,41]. Seminal work by Fink and Cookson[45] detailed osmotic cell lysis following *S. typhimurium* infection, and demonstrated that osmoprotectant agents could block this effect. However, a growing body of literature supports the notion that cell lysis can be separated from inflammasome activation[18]. If lysis can be delayed—which is likely the case in vivo—cells will extrude GSDMD pores using the ESCRT-III machinery and thus survive[24]. Indeed, while culture media carefully balance osmoles across the plasma membrane, no effort is made to balance oncotic pressure. Specifically, large molecules of significant radius to

maintain fluid in the extracellular space, which would be abundant in plasma or tissue fluid, are not present in serum-free medium[46]. Consistent with this notion, the addition of polyethylene glycol of increasing molecular weights prevented lysis in cells overexpressing a constitutively active form of GSDMD[29]. Of critical relevance to the in vivo state are limited reports on children with inborn activating mutations in the gene encoding NLRP3[47]. Such children develop such profound spontaneous inflammasome activation that they are treated with chronic dosing of IL-1 receptor antagonist[48]. Such treatment results in rapid resolution of their profound inflammation. However, there has been no documented leukopenia or increase in replacement of leukocytes from the bone marrow in these patients[48]. If inflammasome activation was a substantial cause of cell death in vivo, one would expect these children to have a substantial loss of leukocytes from circulation, or evidence of their rapid replacement from the marrow. The absence of this finding is inconsistent with a high prevalence of cell death following inflammasome activation in the in vivo state.

It follows that if HMGB1 release requires cell lysis, then our in vitro data add valuable new information to this debate as it

relates to HMGB1 and potentially other DAMPs during systemic inflammatory responses. If cell lysis is not a common feature of inflammasome activation in vivo, that we found no evidence of HMGB1 without cell lysis calls into question whether GSDMD-dependent pyroptosis is an important contributor to HMGB1 secretion in endotoxemia. To that end, we demonstrate in a mouse model that inflammasome activation and HMGB1 release are likely parallel processes. Blockade of IL-1β release in $Gsdmd^{-/-}$ mice, which necessarily blocks osmotic cell lysis, has no effect on HMGB1 secretion in this relevant in vivo model. Given that we employ whole animal GSDMD knockouts, all potential cellular sources of extracellular HMGB1 that are dependent on GSDMD for release would be inhibited. This would include myeloid cells, hepatocytes and other tissue types. The role of GSDMD and pyroptosis in these non-myeloid cell types is of great interest and under active investigation[37]. Taken together, our data provide important mechanistic details to HMGB1 release in the context of inflammation and NLRP3 activation in macrophages. Our results are consistent with the notion that HMGB1 release is a marker of cellular damage, and not necessarily cellular inflammation. Our future work will focus on further defining HMGB1 sources and secretion pathways during systemic inflammation.

In addition to the pyroptosis-mediated mechanism studied here, other cell death pathways, such as necroptosis, ferroptosis, and apoptosis, have also been implicated in HMGB1 release from cells. Necroptosis and ferroptosis are regulated forms of necrosis, distinct from apoptosis and pyroptosis[49,50]. In ferroptosis, HMGB1 can be released in an autophagy-dependent fashion[13]. Recent work using a novel FRET-based necroptosis sensor identified two modes of necroptosis-induced HMGB1 release[14] corresponding to either a rapid burst or a slower sustained release. The differential modes may serve different biological functions and reflect differences in nuclear and plasma membrane damage. The sustained mode of HMGB1 release is likely maintained by activation of ESCRT-III membrane repair machinery[14], which is also activated during pyroptosis to maintain cellular integrity[24]. Whether a similar sustained mode of HMGB1 release occur in pyroptosis is an intriguing prospect that should be evaluated.

Although more extensively studied than necroptosis and ferroptosis, the literature on apoptosis-associated HMGB release is conflicting. Early work suggested that apoptosis limited HMGB1 secretion into the extracellular space[15]. This is consistent with recent data[14,51] but conflicts with a series of studies suggesting that apoptotic caspases and DNAases regulate HMGB1 during apoptosis[52–55]. Further delineation of HMGB1 release during apoptosis is needed given the extensive crosstalk between these various programmed cell death pathways and their executing caspases[56,57]. A more complete characterization of HMGB1 release would inform our understanding of how cell death modulates inflammation, tissue repair, and homeostasis.

## Methods

**Mice**. All animal procedures were approved by the Animal Care Committee at the Hospital for Sick Children and were performed in accordance with regulations and standards from the Animals for Research Act of Ontario and the Canadian Council on Animal Care. Gasdermin D knockout mice ($Gsdmd^{-/-}$, C57Bl/6 background, 8–12 weeks of age) were the kind gift of V. Dixit (Genentech, Inc.) and maintained in pathogen-free conditions with food and water available ad libitum in fully-accredited facilities within the Hospital for Sick Children. Age and sex-matched wild-type littermates were used as control cohorts for in vivo endotoxemia experiments. Additional bone marrow for in vitro experiments was collected from mixed-sex cohorts of wild-type C57Bl/6J mice from Jackson Laboratories.

**Cells**. Primary bone marrow derived macrophages (BMDM) were harvested from the femurs and tibia of wild-type or $Gsdmd^{-/-}$ mice. Cleaned bones were cut and centrifuged to collect bone marrow cells into sterile PBS pH 7.4. The collected bone

marrow suspension was washed with PBS and plated in DMEM supplemented with 10 ng ml$^{-1}$ M-CSF (Peprotech Inc, 315-02). After days 5 of culture the cells were detached from the dishes with TBS with 5 mM EDTA, resuspended in fresh DMEM and plated in 12-well plates.

**Pyroptosis activation and inhibition**. Pyroptosis was activated in macrophages primed with LPS. LPS from E. coli serotype 055:B5 was reconstituted at a stock concentration of 1 mg ml$^{-1}$ and used in assays at a concentration of 100–1000 ng ml$^{-1}$. The pyroptosis pathway was activated using nigericin, 1-palmitoyl-2-glutaryl-sn-glycero-3-phosphocholine (PGPC), or potassium depletion. Potassium-free media (prepared as described in Munoz-Planillo et al.[58]) was added to LPS-primed BMDMs following three washes in potassium-free media for 2 h. Inflammasome activation through bacterial infection was done using the S. aureus strain SA113 ΔoatA[54,55], a generous gift from Dr. Jonathan Kagan (Boston Children's Hospital, Boston, MA, USA). The bacteria were cultured and prepared as described[18]. Briefly, colonies were grown in Todd-Hewitt broth supplemented with Kanamycin (50 μg ml$^{-1}$) for 18 h at 37 °C. Cultures were washed in PBS three times, and resuspended at the desired multiplicity of infection (MOI) in DMEM plus 10% FBS.

Nigericin was purchased from Sigma (N7143), resuspended in ethanol at a stock concentration of 10 mM, and used at a concentration of 20 μM. PGPC was obtained from Avanti Polar Lipids (870602). The chloroform was evaporated using a gentle nitrogen gas stream. Warmed serum-free medium was added to the dried lipid at a final concentration of 1 mg ml$^{-1}$. The medium containing the reconstituted lipids were vortexed for 10 s before being added to the cells at a concentration of 50 or 100 μg ml$^{-1}$. Pyroptosis was inhibited using glycine[30], disulfiram[59,60], or necrosulfonamide[21]. Pneumolysin was obtained from Dr. John Brumell (Hospital for Sick Children, Toronto).

**LDH assay and ELISAs**. For secretion assays, BMDMs were seeded in 12 well plates (200,000 cells per well). The following day the cells were primed or not for 3 h with LPS in regular culture media as indicated in the figure legends. After 3 h the media was changed to 0.1% FBS with or without LPS. Glycine, necrosulfonamide, or disulfiram (as indicated in the figure legends) was then added for 1.5 h (±LPS) prior to the addition or not of nigericin (20 μM) for the final 30 min. For PGPC experiments, after 3 h LPS priming, PGPC was prepared in regular media and incubated with or without LPS for 4 h. At the end of the incubations, cell culture supernatants were collected and cleared of debris by centrifugation at 500 × g for 5 min. The cells were washed 1× with PBS and lysed in lysis buffer provided in the LDH assay kit. Supernatants and lysates were assayed for LDH using an LDH cytotoxity colorimetric assay kit as per the manufacturer's instructions (Pierce, Inc. 88954). IL-1β and HMGB1 levels were quantitatively measured from cell-free culture supernatants by ELISA (IL-1β Abcam ab197742; HMGB1 IBL International ST51011) according to the manufacturers' protocols.

**Secretion assays and western blot analysis**. BMDMs were seeded in 12-well plates as described above. At the end of the incubations, cell culture supernatants were collected and cleared of debris by centrifugation at 500 × g for 5 min. The cells were washed 1× with PBS and lysed in RIPA lysis buffer containing protease inhibitors (Pierce, Protease inhibitor tablet #A32955). Supernatants were tri-chloroacetic acid (TCA) precipitated with 9% TCA on ice and washed with acetone. The precipitates were resuspended in 2× LDS sample buffer and incubated for 5 min at 95 °C. Protein concentration in cell lysates was measured using the DC protein assay kit (Bio Rad). Proteins were resolved using NuPAGE 4–12% Bis-Tris gels (Invitrogen), transferred to PVDF membranes, and immunoblotted with the antibodies indicated in the figure legends according to the manufacturer's recommendations. Antibodies used: HMGB1 (Abcam, ab79823; 1:1000); IL1β (R & D systems, AF-401-NA; 1:1000), pro-caspase 1 + p10 + p12 (Abcam, ab179515;), NLRP3 (AdipoGen, AG-20B-0014); Gsdmd (Santa Cruz, sc-393656), GAPDH (Santa Cruz, sc-25778; 1:1000).

**Fluorescence microscopy**. BMDMs cultured on 18 mm glass coverslips in 12-well plates (150,000 cells per well) and treated as indicated in the figure legends, washed with PBS and fixed in 3% paraformaldehyde at RT for 15 min. The cells were washed three times in PBS, permeabilized with 0.1% TX-100, then incubated in 2% BSA/PBS for 1 h. Primary antibodies were then added for 1 h at RT or overnight at 4 °C. The cells were washed three times with PBS and Cy3 or Alexa488-conjugated secondary antibodies was added for 1 h at room temperature (1:200). The cells were then washed with PBS, incubated for 5 min in 0.5 μM DAPI (Tocris #5748), washed with PBS. Cells were imaged by spinning disk confocal microscopy (Quorum) on a Zeiss Axiovert 200M microscope with a ×63 (or ×25) objectives and an additional ×1.5 magnifying lens. Images were acquired by a CCD camera (Hamamatsu Photonics) driven by the Volocity software. Primary Antibodies used: rabbit anti-human HMGB1 (Abcam ab79823; 1:100) and mouse anti-ASC (Biolegend, Cat: 653902; 1:100).

**Endotoxemia**. Mixed-sex cohorts of 8–10-week-old $Gsdmd^{-/-}$ and wild-type littermate mice were treated with intraperitoneal LPS from E. coli serotype O111:B4 at 20 mg ml$^{-1}$ or saline vehicle. Both wild-type and knockout animals were treated

with the same preparation of LPS or vehicle control. All animals survived to 6 h post-treatment at which point they were humanely euthanized for plasma and tissue collection. All animals were included in the analyses, with the investigators blinded to both genotype and treatment groups. IL-1β and HMGB1 were quantitatively measured from mouse plasma by ELISA (IL-1b abcam ab197742 or HMGB1 Arigo Biolaboratories, ARG81185) according to the manufacturers' protocols.

**Quantification and statistical analysis**. Statistical testing was calculated using Prism 7.0 from GraphPad Software Inc (La Jolla, USA). Experiments with more than two groups were tested using ANOVA with Tukey's multiple comparison test. Unless otherwise specified, the presented data are representative of at least three independent experiments and are provided as mean ± SEM. Post hoc analyses of the in vivo endotoxemia model stratified by sex, did not reveal any evident sex differences in serum IL-1β or HMGB1 levels.

**Reporting summary**. Further information on research design is available in the Nature Research Reporting Summary linked to this article.

## Data availability

The authors declare that the data supporting the findings of this study are available within the article and its Supplementary Information files. Source data are provided with this paper.

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

## Acknowledgements

We thank Dr. V. Dixit (Genentech, Inc.) for providing GSDMD knock-out mice, Dr. Jonathan Kagan (Boston Children's Hospital) for providing the S. aureus (ΔOatA) strain and Dr. John Brumell (The Hospital for Sick Children) for providing recombinant pneumolysin. We thank Dr. Sergio Grinstein (The Hospital for Sick Children) and Dr. Gregory Fairn (St. Michael's Hospital) for comments on the manuscript. This work was funded by the International Anesthesia Research Society Mentored Research Awards (Goldenberg, Steinberg) and The Lung Association—Ontario Grant-in-Aid Program (Steinberg).

## Author contributions

A.V., N.M.G., and B.E.S. conceived the idea of the experiments. Experiments were performed by A.V., A.Y., and L.C. Data were analyzed by A.V., N.M.G., and B.E.S. The manuscript was written by A.V., N.M.G., and B.E.S.

## Competing interests

The authors declare no competing interests.
