## [Peer Review File · Nature Communications]

Reviewers' comments:

Reviewer #1 (Remarks to the Author):

The major claim of the manuscript is that HMGB1 release from macrophages is due to Gasdermin D and inflammasome activation. As such the manuscript ignores critical findings related to the role of autophagy in HMGB1 release (The EMBO Journal (2011) 30, 4701–4711), the increased HMGB1 release in vivo when it is knocked out of macrophages (Proc Natl Acad Sci U S A. 2013 Dec 17;110(51):20699-704.) and its apparent protective role, and the finding that HMGB1 is largely released from hepatocytes during endotoxin stimulation in vivo (J Leukoc Biol. 2019 Jul;106(1):161-169. doi: 10.1002/JLB.3MIR1218-497R.).

The data provided are of interest and the manuscript lucid and the figures clear and interesting. Several critical issues are noted below:

- 1) The authors throughout the ms use the terms alarmin and danger associated molecular pattern molecule. Alarmins technically have intrinsic dendritic cell maturation AND chemotactic activity; Damage Associated Molecular Pattern Molecules or DAMPs is the correct term and should be used throughout.
- 2) All of the assays reported are relatively short term (5hours of LPS stimulation); what happens with longer time intervals and increased concentrations of LPS?
- 3) In the introduction, first paragraph it would be better to comment on the disparate nuclear, cytosolic, and extracellular roles of HMGB1.
- 4) Introduction, third paragraph, the Nature July 11, 2002 paper of Bianchi should be cited demonstrating that HMGB1 is NOT released from apoptotic cells even when strong detergents are added.
- 5) References 10 and 11 do not refer to autophagy; the EMBO JI paper above should be cited.
- 6) Supplemental Figure 1A and B are just lower magnification views of Figures 1A and B and can be eliminated (cited in text).
- 7) Figure 2C, the ASC specks are not well visualized in my print copy, whereas visible in later Figures.
- 8) A positive control (? freeze thaw lysis) showing that LDH is released from Gsdmd ko cells would be more convincing.
- 9) No comments on necroptosis, ferroptosis, and apoptosis and release are presented; this would increase interest in the ms.

Reviewer #2 (Remarks to the Author):

Nature Communications Review Volchuk et al., 2020

In the manuscript entitled "Indirect Regulation of HMGB1 Release By Gasdermin D" the authors clarify the literature surrounding HMGB1 release during inflammation. The authors refute prior literature stating that LPS treatment alone can result in significant HMGB1 release from macrophages and instead posit a model for lytic release of HMGB1 downstream of gasdermin D pore formation in cells with active inflammasomes in vitro. Moreover, the authors demonstrate sufficiency for cellular lysis as a mode of HMGB1 egress through the use of a bacterial pore forming toxin and utilize models of non-lytic inflammasome activation to demonstrate that inflammasome activation is not sufficient for HMGB1 release in the absence of lysis. These findings reinforce the notion that gasdermin D pores are sufficient to release the smaller and non-tethered molecule IL-1beta in situations of lytic and non-lytic inflammasome signaling, while clarifying that the larger and potentially DNA bound molecule HMGB1 may require lysis of any form to be released.

Major Comments:

1. In Figure 4, the authors must conduct a time course analysis of release of LDH, HMGB1, and IL-1 β to confirm or refute their claims that HMGB1 is released via gasdermin D but indirectly (not through the pore) via lysis. It stands to reason that glycine buffering prevents the rapid dissolution of membranes caused during lysis while leaving the gasdermin D perforated membranes intact, thus the kinetics of release of intracellular components may differ between a burst during lysis and a diffusion limited process during glycine treatment. To this effect, DiPeso et al 2017 demonstrate that kinetics of release lag when lysis is prevented for GFP or larger tdTomato molecules, but that eventually even larger molecules like tdTomato are secreted across these perforated membranes during glycine treatment. The same could well hold true for HMGB1 in the context of experimentally preventing lysis with glycine with an earlier time point showing a lag of HMGB1 release that catches up over time. The western blot in Figure 4b suggests that there is some release of HMGB1 even with glycine treatment. The authors should include ELISAs of HMGB1 and IL-1 β overtime after glycine buffering.
2. In Figure 4, the authors should include HMGB1 ELISA data for infection with the mutant *S. aureus* strain.
3. In Figure 5, the authors should utilize an additional physiologic pathway of causing lysis in macrophages by inducing necroptosis to reinforce their conclusions drawn from pyroptotic inflammasome activation and bacterial pore forming toxin treatment. This pathway can be readily induced (Kaiser et al, 2013) through inhibition of caspase-8 (through the pan-caspase inhibitor zVAD-fmk) in combination with LPS for TRIF dependent necroptosis or through other bacterial TLR ligands or recombinant TNF α for a TNFR dependent necroptosis. Again please provide ELISA data for HMGB1 and IL-1 β in addition to the current western blot on supernatants.
4. In Figure 6, the endotoxemia model employed displays rather modest increases in HMGB1. Is this increase from 10 ng/ml to 15 ng/ml HMGB1 physiologically relevant? A more aggressive endotoxemia model such as two doses of LPS injected as opposed to the current single dose may yield larger delta between saline treatment and LPS treatment for IL-1 β and HMGB1 release. Measuring of LDH release into the serum has recently been used in the context of necroptosis and should also be possible during endotoxemia to confirm or refute the authors' assertion that this gasdermin D independent increase in HMGB1 is independent of cellular lysis in vivo (Yoon et al, 2017).

Minor Comments:

1. In Figure 1, does LPS treatment alone globally alter the expression level of HMGB1? Can the authors quantify HMGB1 protein from lysed untreated and LPS treated BMDMs by both western blot and ELISA? An alternative approach may be to utilize your staining protocol for HMGB1 for immunofluorescence microscopy to prepare single cell suspensions to quantify whole cell MFI for HMGB1 signal on a flow cytometer for many thousands of stained cells for untreated and LPS treated conditions.
2. In the section entitled " Pyroptosis activators lead to HMGB1 release from macrophages", a sentence reads "In comparison, LPS treatment alone caused release of LDH, comparable to that of control untreated cells (Figure 2B)" that seems to mean to this reviewer that LPS alone can induce LDH release whereas I believe the authors are trying to contrast the small or lack of release in untreated and LPS cells compared to the inflammasome activated cells that release significant quantities of LDH. Please revise the language to be more clear that LPS and untreated cells do NOT release LDH whereas pyroptotic (lysed) cells do release LDH.
3. In Figure legends time points and concentrations of ligands used should be stated unless all experiments were identical throughout the paper and included in the materials and methods. Figure 4 does not describe the length of time for nigericin treatment when measuring LDH release, HMGB1 release, and IL-1 β release with glycine buffering or NSA treatment.
4. Check over text for inadvertent typos such as the use of the word "effecting" in Figure 4 that should be "affecting".
5. In Figure 5, how does treatment with the bacterial pore forming toxin PL result in nuclear membrane permeabilization to allow for HMGB1 release? Do the authors note activation of

apoptotic caspases after treatment of with PL? Is nuclear membrane dissolution also required for HMGB1 release in addition to plasma membrane perforation / lysis? Is DNA cleavage necessary for HMGB1 release as HMGB1 is bound to nuclear DNA that would occur during pyroptosis or apoptosis?

6. Description of the reagent Δ oatA *S. aureus* in the methods section entitled "Pyroptosis activation and inhibition" should not be described as "pyroptosis activation by bacterial infection" as this stimulation is considered even by the current authors to be a non-lytic (non-pyroptotic) stimulation. The authors could use the term inflammasome activation generally for the section header and use the terms non-lytic inflammasome activation or the recently coined cellular state "hyperactivation" to describe these specific reagents and experiments.

REVIEWER #1

The major claim of the manuscript is that HMGB1 release from macrophages is due to Gasdermin D and inflammasome activation. As such the manuscript ignores critical findings related to the role of autophagy in HMGB1 release (The EMBO Journal (2011) 30, 4701–4711), the increased HMGB1 release in vivo when it is knocked out of macrophages (Proc Natl Acad Sci U S A. 2013 Dec 17;110(51):20699-704.) and its apparent protective role, and the finding that HMGB1 is largely released from hepatocytes during endotoxin stimulation in vivo (J Leukoc Biol. 2019 Jul;106(1):161-169. doi: 10.1002/JLB.3MIR1218-497R.).

Thank you for alerting us to these critical findings, which we have now incorporated into the Discussion section of our revised manuscript. We have also added a more thorough discussion of HMGB1 release in the context of necroptosis, ferroptosis, and apoptosis (point #9 below). The specific changes as suggested by Reviewer #1 are as follows:

First, in our manuscript, we propose that when the NLRP3 inflammasome is activated in macrophages, HMGB1 release is gasdermin D-dependent but does not traverse the gasdermin D pore. Instead, under these conditions, HMGB1 release requires pyroptotic cell death. Our revised manuscript now emphasizes our focus on the mechanism of HMGB1 release to the extracellular space during NLRP3 activation. Our original manuscript did not clearly outline this as our central claim, which could result in confusion. This has been clarified in multiple sections, including the Abstract and Discussion:

Abstract:

“Together, these data demonstrate that *in vitro* HMGB1 release following inflammasome activation occurs following cellular rupture, which is likely inflammasome-independent *in vivo*.”

Discussion:

“Taken together, our data provide important mechanistic details to HMGB1 release in the context of inflammation and NLRP3 activation in macrophages. Our results are consistent with the notion that HMGB1 release is a marker of cellular damage, and not necessarily cellular inflammation. Our future work will focus on further defining HMGB1 sources and secretion pathways during systemic inflammation.”

Second, other HMGB1 release mechanisms outlined by Reviewer #1, such as an autophagy-mediated process, are possible sources of extracellular HMGB1. We thank the Reviewer for pointing this out and providing direction on other HMGB1 secretion mechanisms, which are now referenced. Our data do not preclude an interaction between cytosolic HMGB1 and autophagy or other cell death pathways, which may provide additional routes for HMGB1 release. This is summarized in our revised Discussion section.

“... While GSDMD pores are theoretically large enough to allow HMGB1 egress, HMGB1 may exist in a multimeric state and/or bound to other proteins or nucleic acids, largely increasing its effective hydrodynamic radius.

Together, these *in vitro* data suggest that when the NLRP3 inflammasome is activated in macrophages, HMGB1 release is gasdermin D-dependent but does not traverse the gasdermin D pore. Instead, under these conditions, HMGB1 release requires pyroptotic cell death. Notably, our results do not preclude other active HMGB1 secretion pathways. For example, cellular oxidative stress drives Atg5-dependent autophagy-mediated HMGB1 release.^{1,2} This unconventional secretion pathway is notable in that in a reciprocal relationship, cytosolic HMGB1 has been shown to be protective against endotoxemia and bacterial infection via its interaction with the autophagy pathway.³ By activating autophagy, HMGB1 inhibits inflammasomes, consistent with reports that basal autophagy can suppress IL-1 β secretion.

The distinction between direct egress through the GSDMD pores versus cell membrane rupture has important implications to *in vivo* HMGB1 physiology. ...”

Lastly, we thank Reviewer #1 for highlighting that *in vivo* data do not show diminished serum HMGB1 in endotoxemia when HMGB1 is knocked out of myeloid cells,³ indicating a non-macrophage source. As noted, hepatocytes are the likely source of circulating HMGB1 in endotoxemia.⁴ We have referenced these important findings in our revised manuscript.

Our *in vivo* experiments employ animals completely devoid of gasdermin D. As such, all potential cellular sources of extracellular HMGB1 that are dependent on gasdermin D for release would be inhibited. This would include myeloid cells, hepatocytes and other tissue types. While our *in vitro* work investigates signaling in macrophages, we did not intend to propose that they are the only source of HMGB1 in endotoxemia. As noted above, hepatocytes are likely major contributors to circulating HMGB1 in endotoxemia. Macrophages, however, are known to release HMGB1 upon LPS-stimulation and so we used them as target tissue for mechanistic investigation of gasdermin D in HMGB1 release. Our updated manuscript clarifies these sources of confusion and reference the above findings:

“To test the hypothesis that HMGB1 release and pyroptotic cell death are spatially unrelated in endotoxemia, we used an acute intraperitoneal LPS injection model in wild-type and GSDMD^{-/-} mice. In this model, **serum HMGB1 levels increase likely from a hepatocyte source^{4,5} and remain elevated along with other pro-inflammatory cytokines including IL-1 β within hours of LPS administration.** Mice were injected with 20 mg/kg LPS or vehicle, and were sacrificed 6 hours later.”

“To that end, we demonstrate in a mouse model that inflammasome activation and HMGB1 release are likely parallel processes. Blockade of IL-1 β release in GSDMD^{-/-} mice, which necessarily blocks osmotic cell lysis, has no effect on HMGB1 secretion in this relevant *in vivo* model. **Given that we employ whole animal gasdermin D knockouts, all potential cellular sources of extracellular HMGB1 that are dependent on gasdermin D for release would be inhibited. This would include myeloid cells and other tissue types.** Taken together, our data provide important mechanistic details to HMGB1 release in the context of inflammation.”

1) The authors throughout the ms use the terms alarmin and danger associated molecular pattern molecule. Alarmins technically have intrinsic dendritic cell maturation AND chemotactic activity; Damage Associated Molecular Pattern Molecules or DAMPs is the correct term and should be used throughout.

Thank you for suggesting this important clarification to the nomenclature. We have amended all such mentions to “Damage Associated Molecular Patterns (DAMP)”.

2) All of the assays reported are relatively short term (5hours of LPS stimulation); what happens with longer time intervals and increased concentrations of LPS?

Thank you for this question. Indeed, previous reports have employed variable concentrations and durations of LPS stimulation in the assessment of HMGB1 release, and our standard regimen of 0.5 μ g/ml for 5 hours is on the higher end of doses used for initiation of pyroptosis. We have completed several experiments with varying doses and durations of LPS treatment using the

human monocytic leukemia cell line, THP-1, differentiated overnight in phorbol 12-myristate 13-acetate (PMA). Following treatment with LPS 0.5 ug/ml for 5 hours, or 0.2 ug/ml for 24 hours, we observed no release of HMGB1 into the extracellular medium (Figure 1A). The addition of nigericin, 20 uM for 30 minutes caused substantial HMGB1 release, however simultaneous release of GAPDH into the medium was also observed. GAPDH as a tetramer is too large to traverse a gasdermin D pore, therefore it is likely that HMGB1 release in this context occurs during cellular rupture. This hypothesis is consistent with our LDH release data, demonstrating significant cytotoxicity as a result of this regimen (Figure 1B). That 24 hour LPS priming results in less cytotoxicity following nigericin stimulation is consistent with previous reports demonstrating a downregulation of inflammasome components following prolonged LPS treatment.⁶ Together, these data suggest that HMGB1 release occurs concurrently with cell lysis over a range of LPS doses and treatment durations.

3) In the introduction, first paragraph it would be better to comment on the disparate nuclear, cytosolic, and extracellular roles of HMGB1.

Thank you for this important suggestion. We have now added discussion to the first paragraph of the introduction relating to the roles of HMGB1 in the nucleus, cytosol, and extracellular space:

High-mobility group box 1 (HMGB1) is a ubiquitously expressed protein that can be released from cells during infection or sterile inflammation.⁷ **HMGB1 can exist in several spatial pools, with the bulk of HMGB1 in a resting cell residing in the nucleus. HMGB1 weakly binds to chromosomal DNA and can influence DNA transcription through the regulation of chromatin structure.⁸ Indeed, loss of HMGB1 from macrophage results in nuclear reprogramming toward a more activated state.³ Multiple stimuli can lead to HMGB1 translocation to a cytosolic pool.⁷ This is thought to result from acetylation of two nuclear localization sequences on HMGB1.⁹ Additionally, phosphorylation of cytosolic HMGB1 can inhibit its import to the nucleus, shifting its steady state toward the cytosol.¹⁰** Bacterial products, such as lipopolysaccharide (LPS), and tissue injury can stimulate the release of HMGB1 from both immune and parenchymal cells.^{7,11} Once in the extracellular space, HMGB1 is a potent signaling molecule. The binding of HMGB1 to its cognate receptors – **including the receptor for advanced end-glycation products (RAGE), toll-like receptor 4 (TLR4) and others** - stimulates cytokine secretion from macrophages, and can also promote proliferation, migration, and other phenotypic changes in somatic cells.¹²⁻¹⁴ Its pleiotropic effects make extracellular HMGB1 the focus of translational research in a wide variety of fields ranging from sepsis, acute respiratory distress syndrome, inflammatory arthritis, and pulmonary vascular diseases.^{7,11,15}

4) Introduction, third paragraph, the Nature July 11, 2002 paper of Bianchi should be cited demonstrating that HMGB1 is NOT released from apoptotic cells even when strong detergents are added.

Thank you for your suggestion to highlight this important work. The updated manuscript has now incorporated this reference.

5) References 10 and 11 do not refer to autophagy; the EMBO JI paper above should be cited.

Thank you for identifying this issue. We have removed those two references and replaced them with the EMBO J paper suggested.

6) Supplemental Figure 1A and B are just lower magnification views of Figures 1A and B and can be eliminated (cited in text).

We have removed this Supplemental Figure, and believe it helps streamline our overall submission. Thank you for this suggestion.

7) Figure 2C, the ASC specks are not well visualized in my print copy, whereas visible in later Figures.

We have adjusted the contrast of the ASC channel evenly across these panels in order to enhance their visualization in print.

8) A positive control (? freeze thaw lysis) showing that LDH is released from Gsdmd ko cells would be more convincing.

Thank you for this suggestion. Indeed, we have not explored potential conditions under which GSDMD knockout cells would rupture. Your suggestion of freeze/thaw cycles would be expected to yield significant LDH release from these cells, since this process should not be gasdermin D-dependent then we would expect HMGB1 to be released concurrently. Given our government-mandated laboratory closure, we cannot currently carry out this experiment.

9) No comments on necroptosis, ferroptosis, and apoptosis and release are presented; this would increase interest in the ms.

Thank you for this suggestion. We agree that further discussion around HMGB1 release in the context of necroptosis, ferroptosis and apoptosis would increase interest in the manuscript. To that end, we have added the following text to our Discussion:

“In addition to the pyroptosis-mediated mechanism studied here, other cell death pathways, such as necroptosis, ferroptosis and apoptosis, have also been implicated in HMGB1 release from cells. Necroptosis and ferroptosis are regulated forms of necrosis, distinct from apoptosis and pyroptosis.^{16,17} In ferroptosis, HMGB1 can be released in an autophagy-dependent fashion.¹⁸ Recent work using a novel FRET-based necroptosis sensor identified two modes of necroptosis-induced HMGB1 release¹⁹ corresponding to either a rapid burst or a slower sustained release. The differential modes may serve different biological functions and reflect differences in nuclear and plasma membrane damage. The sustained mode of HMGB1 release is likely maintained by activation of ESCRT-III membrane repair machinery,¹⁹ which is also activated during pyroptosis to maintain cellular integrity.²⁰ Whether a similar sustained mode of HMGB1 release occurs in pyroptosis is an intriguing prospect that should be evaluated.

Although more extensively studied than necroptosis and ferroptosis, the literature on apoptosis-associated HMGB release is conflicting. Early work suggested that apoptosis limited HMGB1 secretion into the extracellular space.²¹ This is consistent with recent studies^{19,22} but conflicts with a series of studies suggesting that apoptotic caspases and DNAses regulate HMGB1 during apoptosis.^{23–26} Further delineation of HMGB1 release during apoptosis is needed given the extensive crosstalk between these various programmed cell death pathways and their executing caspases.^{27,28} A more complete characterization of HMGB1 release would inform our understanding of how cell death modulates inflammation, tissue repair, and homeostasis.”

REVIEWER #2

1. In Figure 4, the authors must conduct a time course analysis of release of LDH, HMGB1, and IL-1 β to confirm or refute their claims that HMGB1 is released via gasdermin D but indirectly (not through the pore) via lysis. It stands to reason that glycine buffering prevents the rapid dissolution of membranes caused during lysis while leaving the gasdermin D perforated membranes intact, thus the kinetics of release of intracellular components may differ between a burst during lysis and a diffusion limited process during glycine treatment. To this effect, DiPeso et al 2017 demonstrate that kinetics of release lag when lysis is prevented for GFP or larger tdTomato molecules, but that eventually even larger molecules like tdTomato are secreted across these perforated membranes during glycine treatment. The same could well hold true for HMGB1 in the context of experimentally preventing lysis with glycine with an earlier time point showing a lag of HMGB1 release that catches up over time. The western blot in Figure 4b suggests that there is some release of HMGB1 even with glycine treatment. The authors should include ELISAs of HMGB1 and IL-1 β overtime after glycine buffering.

Thank you for this important insight. Indeed, the kinetics of release from a “semi-intact” cell, or a cell undergoing a slower lytic process, will likely depend upon the speed of this process as well as the size and mobility of the molecule of interest. To examine this potential effect, and whether HMGB1 release is simply delayed by glycine in the context of inflammasome activation, we were able to perform a time course experiment, as suggested by the reviewer. Mouse BMDM were treated with LPS (0.5 μ g/ml) for a total of 5 hours. Nigericin, with or without glycine, was added to this solution for the times indicated (Figure 2), ranging from 15 to 90 minutes. By LDH release, nigericin caused a time-dependent increase in cytotoxicity, which was effectively blocked by the addition of glycine. There was a very modest loss in this protective effect at 90 minutes, but even at this long duration a dramatic glycine effect was observed. Glycine showed a strong protective effect against HMGB1 release across all time points, supporting the notion that HMGB1 largely remains inside an intact cell, in spite of gasdermin D pore formation, at least up to 90 minutes after inflammasome activation. These data strengthen our results, and demonstrate a temporal durability to our observation. Thank you for suggesting this important experiment.

2. In Figure 4, the authors should include HMGB1 ELISA data for infection with the mutant S. aureus strain.

Thank you very much for this suggestion. Unfortunately, without access to our laboratory for an indefinite period we are unable to complete this experiment at this time. Our semi-quantitative western blot data consistently demonstrate the effects outlined in our manuscript, and have been quantified and statistically analyzed for significance.

3. In Figure 5, the authors should utilize an additional physiologic pathway of causing lysis in macrophages by inducing necroptosis to reinforce their conclusions drawn from pyroptotic inflammasome activation and bacterial pore forming toxin treatment. This pathway can be readily induced (Kaiser et al, 2013) through inhibition of caspase-8 (through the pan-caspase inhibitor zVAD-fmk) in combination with LPS for TRIF dependent necroptosis or through other bacterial TLR ligands or recombinant TNF α for a TNFR dependent necroptosis. Again please provide ELISA data for HMGB1 and IL-1 β in addition to the current western blot on supernatants.

Thank you for this important insight. It would be fascinating to investigate the release of HMGB1 from immune cells during other processes, including necroptosis, ferroptosis, and other forms of regulated cell death. We cannot currently complete these experiments, but hope to pursue a broader survey of pathways that do and do not lead to HMGB1 release in our future studies. As of now, we cannot comment on the importance of necroptosis with respect to HMGB1 release in our system. Our revised manuscript now includes discussion of HMGB1 release in necroptosis in the Discussion.

4. In Figure 6, the endotoxemia model employed displays rather modest increases in HMGB1. Is this increase from 10 ng/ml to 15 ng/ml HMGB1 physiologically relevant? A more aggressive endotoxemia model such as two doses of LPS injected as opposed to the current single dose may yield larger delta between saline treatment and LPS treatment for IL-1 β and HMGB1 release. Measuring of LDH release into the serum has recently been used in the context of necroptosis and should also be possible during endotoxemia to confirm or refute the authors' assertion that this gasdermin D independent increase in HMGB1 is independent of cellular lysis in vivo (Yoon et al, 2017).

This is an important question, and we thank you for raising it. Models for *in vivo* release demonstrate a high rate of variability with respect to the concentration of plasma HMGB1. Our increase of approximately 5 ng/ml is within range of the reported dissociation constant for HMGB1-TLR4 interaction (0.4 μ M).²⁹ Indeed, HMGB1 has potent bioactivity in the extracellular space. Furthermore, while assessing circulating plasma concentrations, we do not know what the actual concentration of HMGB1 is in tissues, which are the actual effect site. It is possible that plasma measurements largely underestimate the amount of HMGB1 found locally within a target tissue. Unfortunately due to our laboratory closure we are currently unable to attempt to repeat this experiment with a larger LPS dose for the foreseeable future.

Minor Comments:

1. In Figure 1, does LPS treatment alone globally alter the expression level of HMGB1? Can the authors quantify HMGB1 protein from lysed untreated and LPS treated BMDMs by both western blot and ELISA? An alternative approach may be to utilize your staining protocol for HMGB1 for immunofluorescence microscopy to prepare single cell suspensions to quantify whole cell

MFI for HMGB1 signal on a flow cytometer for many thousands of stained cells for untreated and LPS treated conditions.

We appreciate this important suggestion. We have now included data regarding the expression of HMGB1 in BMDM cell lysates from control and LPS-treated cells as Figure 1B in the revised manuscript. We do not observe any significant up- or down-regulation of HMGB1 in response to LPS treatment.

2. In the section entitled “ Pyroptosis activators lead to HMGB1 release from macrophages”, a sentence reads “In comparison, LPS treatment alone caused release of LDH, comparable to that of control untreated cells (Figure 2B)” that seems to mean to this reviewer that LPS alone can induce LDH release whereas I believe the authors are trying to contrast the small or lack of release in untreated and LPS cells compared to the inflammasome activated cells that release significant quantities of LDH. Please revise the language to be more clear that LPS and untreated cells do NOT release LDH whereas pyroptotic (lysed) cells do release LDH.

Thank you for pointing out this inconsistency in our writing. We have amended the sentence at issue to clarify this finding.

3. In Figure legends time points and concentrations of ligands used should be stated unless all experiments were identical throughout the paper and included in the materials and methods. Figure 4 does not describe the length of time for nigericin treatment when measuring LDH release, HMGB1 release, and IL-1 β release with glycine buffering or NSA treatment.

Thank you again for noting this. We have now added specific experimental details for all treatments to our figure legends.

REFERENCES

1. Tang, D. *et al.* Endogenous HMGB1 regulates autophagy. *J. Cell Biol.* **190**, 881–892 (2010).
2. Dupont, N. *et al.* Autophagy-based unconventional secretory pathway for extracellular delivery of IL-1 β . *EMBO J.* **30**, 4701–4711 (2011).
3. Yanai, H. *et al.* Conditional ablation of HMGB1 in mice reveals its protective function against endotoxemia and bacterial infection. *Proc. Natl. Acad. Sci.* **110**, 20699–20704 (2013).
4. Deng, M., Scott, M. J., Fan, J. & Billiar, T. R. Location is the key to function: HMGB1 in sepsis and trauma-induced inflammation. *J. Leukoc. Biol.* **106**, 161–169 (2019).
5. Deng, M. *et al.* The Endotoxin Delivery Protein HMGB1 Mediates Caspase-11-Dependent Lethality in Sepsis. *Immunity* (2018) doi:10.1016/j.immuni.2018.08.016.
6. Hu, Y. *et al.* Tripartite-Motif Protein 30 Negatively Regulates NLRP3 Inflammasome Activation by Modulating Reactive Oxygen Species Production. *J. Immunol.* **185**, 7699–7705 (2010).
7. Andersson, U. & Tracey, K. J. HMGB1 is a therapeutic target for sterile inflammation and infection. *Annu. Rev. Immunol.* **29**, 139–162 (2011).

8. Ueda, T. & Yoshida, M. HMGB proteins and transcriptional regulation. *Biochim. Biophys. Acta BBA - Gene Regul. Mech.* **1799**, 114–118 (2010).
9. Bonaldi, T. *et al.* Monocytic cells hyperacetylate chromatin protein HMGB1 to redirect it towards secretion. *EMBO J.* **22**, 5551–5560 (2003).
10. Youn, J. H. & Shin, J. S. Nucleocytoplasmic Shuttling of HMGB1 Is Regulated by Phosphorylation That Redirects It toward Secretion. *J. Immunol.* **177**, 7889–7897 (2006).
11. Harris, H. E., Andersson, U. & Pisetsky, D. S. HMGB1: a multifunctional alarmin driving autoimmune and inflammatory disease. *Nat. Rev. Rheumatol.* **8**, 195–202 (2012).
12. Fiuza, C. *et al.* Inflammation-promoting activity of HMGB1 on human microvascular endothelial cells. *Blood* **101**, 2652–2660 (2003).
13. Goldenberg, N. M. *et al.* Therapeutic Targeting of High-Mobility Group Box-1 in Pulmonary Arterial Hypertension. *Am. J. Respir. Crit. Care Med.* **199**, 1566–1569 (2019).
14. Wang, H. *et al.* HMG-1 as a late mediator of endotoxin lethality in mice. *Science* **285**, 248–251 (1999).
15. Angus, D. C. *et al.* Circulating high-mobility group box 1 (HMGB1) concentrations are elevated in both uncomplicated pneumonia and pneumonia with severe sepsis*. *Crit. Care Med.* **35**, 1061–1067 (2007).
16. Xie, Y. *et al.* Ferroptosis: process and function. *Cell Death Differ.* **23**, 369–379 (2016).
17. Pasparakis, M. & Vandenabeele, P. Necroptosis and its role in inflammation. *Nature* **517**, 311–320 (2015).
18. Wen, Q., Liu, J., Kang, R., Zhou, B. & Tang, D. The release and activity of HMGB1 in ferroptosis. *Biochem. Biophys. Res. Commun.* **510**, 278–283 (2019).
19. Murai, S. *et al.* A FRET biosensor for necroptosis uncovers two different modes of the release of DAMPs. *Nat. Commun.* **9**, 4457 (2018).
20. Rühl, S. *et al.* ESCRT-dependent membrane repair negatively regulates pyroptosis downstream of GSDMD activation. *Science* **362**, 956–960 (2018).
21. Scaffidi, P., Misteli, T. & Bianchi, M. E. Release of chromatin protein HMGB1 by necrotic cells triggers inflammation. *Nature* **418**, 191–195 (2002).
22. Murakami, Y. *et al.* Programmed necrosis, not apoptosis, is a key mediator of cell loss and DAMP-mediated inflammation in dsRNA-induced retinal degeneration. *Cell Death Differ.* **21**, 270–277 (2014).
23. Bell, C. W., Jiang, W., Reich, C. F. & Pisetsky, D. S. The extracellular release of HMGB1 during apoptotic cell death. *Am. J. Physiol. Cell Physiol.* **291**, C1318–1325 (2006).
24. Kazama, H. *et al.* IMMUNE TOLERANCE INDUCTION BY APOPTOTIC CELLS REQUIRES CASPASE-DEPENDENT OXIDATION OF HMGB1. *Immunity* **29**, 21–32 (2008).
25. Yamada, Y. *et al.* DR396, an apoptotic DNase γ inhibitor, attenuates high mobility group box 1 release from apoptotic cells. *Bioorg. Med. Chem.* **19**, 168–171 (2011).
26. Yamada, Y. *et al.* The release of high mobility group box 1 in apoptosis is triggered by nucleosomal DNA fragmentation. *Arch. Biochem. Biophys.* **506**, 188–193 (2011).
27. Legrand, A. J., Konstantinou, M., Goode, E. F. & Meier, P. The Diversification of Cell Death and Immunity: Memento Mori. *Mol. Cell* **76**, 232–242 (2019).
28. Van Opdenbosch, N. & Lamkanfi, M. Caspases in Cell Death, Inflammation, and Disease. *Immunity* **50**, 1352–1364 (2019).

29. He, M., Bianchi, M. E., Coleman, T. R., Tracey, K. J. & Al-Abed, Y. Exploring the biological functional mechanism of the HMGB1/TLR4/MD-2 complex by surface plasmon resonance. *Mol. Med.* **24**, 21 (2018).

A

B

LPS 0.5 μ g/ml, 5 hr

LPS 0.2 μ g/ml, 24 hr

Nigericin 20 μ M, 30 min

-	+	+	-	-
-	-	-	+	+
-	-	+	-	+

Figure 1: HMGB1 release from human THP-1 cells following LPS treatment. THP-1 cells, differentiated overnight in phorbol 12-myristate 13-acetate (PMA), were treated with LPS at the indicated doses and durations. Nigericin (Ng) was added at 20 μ M for 30 minutes. **A.** Cells supernatants were collected, TCA precipitated and proteins were resolved by SDS-PAGE and western blotted for HMGB1 or GAPDH. Untreated (Ctrl) cells, cells treated with LPS alone (LPS), or LPS + Ng. **B.** Colorimetric LDH release assays was performed on cell culture supernatants from (A) and from the cell lysates. LDH release was quantitated as a percentage LDH in the supernatant to total LDH in lysed cells. Data are shown as mean \pm SEM, n = 4 independent experiments.

A

B

Figure 2: Time course of HMGB1 release in response to increasing duration of nigericin treatment in absence or presence of glycine. Primary mouse BMDM were primed with LPS (0.5 μg/ml) for a total of 5 hours or left untreated (Ctrl). Nigericin (20 μM) was added during this priming period for the indicated duration (15-90 min). In specified samples, glycine (5 mM) was added for the final 2 hours of the LPS prime. **A.** Cell culture supernatants were collected and assayed by western blot for HMGB1, GAPDH, and processed IL-1β. **B.** Supernatants from **A.** were assessed for LDH release by colorimetric assay. Data are expressed as percent of total LDH in lysates. Note that glycine is at least partially effective in preventing cytotoxicity over the entire tested time frame, and that HMGB1 release follows the trend of lysis at all tested time points.

REVIEWERS' COMMENTS:

Reviewer #2 (Remarks to the Author):

In this revision, my concerns have been addressed. The authors improved what was already a solid study, which should be well-received by the community.

Reviewer #3 (Remarks to the Author):

The authors have satisfied most of the comments raised by reviewer 1 in their revised paper. In terms of comment #2 from reviewer 1 (and to some extent comment #4 from Reviewer 2) the issue of the experiment looking at LPS concentrations and time courses remains unaddressed. This is an important question if the authors want to conclude that HMGB1 release is completely and always GSDMD-independent in vivo. This conclusion is based on a single time point (6 hrs) and LPS dose in vivo. There is no mention of the potential roles of caspase-11, the intracellular LPS receptor, as a path to GSDMD cleavage and potential HMGB1 release. This would probably only be observed at later time points (as caspase-11 requires time to be unregulated) and would be amplified by LPS pre-treatment as suggested by Reviewer #2. This might also be seen in vitro at higher LPS concentrations and at later time points (due to proptosis). If the authors cannot carry out time courses and dose responses, I believe that authors should mention the roles of caspase-11 in LPS responses and narrow their conclusion to indicate that the early elevations on circulating HMGB1 levels in vivo seem to be GSDMD-independent and that they cannot exclude a role for caspase-11 driven GSDMD cleavage at later time points.

REVIEWER #2

In this revision, my concerns have been addressed. The authors improved what was already a solid study, which should be well-received by the community.

Thank you very much for your review of our manuscript. We greatly appreciate your comments, which have resulted in what we believe to be a much-improved final version.

REVIEWER #3

The authors have satisfied most of the comments raised by reviewer 1 in their revised paper. In terms of comment #2 from reviewer 1 (and to some extent comment #4 from Reviewer 2) the issue of the experiment looking at LPS concentrations and time courses remains unaddressed. This is an important question if the authors want to conclude that HMGB1 release is completely and always GSDMD-independent in vivo. This conclusion is based on a single time point (6 hrs) and LPS dose in vivo. There is no mention of the potential roles of caspase-11, the intracellular LPS receptor, as a path to GSDMD cleavage and potential HMGB1 release. This would probably only be observed at later time points (as caspase-11 requires time to be unregulated) and would be amplified by LPS pre-treatment as suggested by Reviewer #2. This might also be seen in vitro at higher LPS concentrations and at later time points (due to proptosis). If the authors cannot carry out time courses and dose responses, I believe that authors should mention the roles of caspase-11 in LPS responses and narrow their conclusion to indicate that the early elevations on circulating HMGB1 levels in vivo seem to be GSDMD-independent and that they cannot exclude a role for caspase-11 driven GSDMD cleavage at later time points.

We thank you for this important insight into our work. Indeed, the possibility exists that under different conditions, including longer LPS exposures, or higher doses of LPS, non-canonical inflammasome pathways may be activated that result in HMGB1 release, both *in vivo* and *in vitro*. Unfortunately, the government-mandated shut down of our laboratory necessitated the reduction of our animal colonies down to several breeders. As such, we are unable to carry out further time course experiments in a timely fashion at this point. In lieu of these further experiments, we have taken your suggestion of modifying our conclusions to account for the concerns you have correctly raised. A new section of the discussion now reads:

Of note, it remains possible that longer durations of LPS exposure, both in vitro and in vivo, may reveal HMGB1 release through the GSDMD pore. This is especially interesting given the potential role for caspase 11-dependent, non-canonical inflammasome-mediated routes of secretion. As an intracellular LPS sensor, this complex may be involved in directing HMGB1 secretion over longer periods of LPS priming, and will be the subject of future investigations.